# TFC-GCN: Lightweight Temporal Feature Cross-Extraction Graph Convolutional Network for Skeleton-Based Action Recognition

**DOI:** 10.3390/s23125593

**Published:** 2023-06-15

**Authors:** Kaixuan Wang, Hongmin Deng

**Affiliations:** College of Electronics and Information Engineering, Sichuan University, No. 24, Section 1, First Ring Road, Wuhou District, Chengdu 610041, China; 2021222050059@stu.scu.edu.cn

**Keywords:** deep learning, action recognition, graph convolutional networks, lightweight

## Abstract

For skeleton-based action recognition, graph convolutional networks (GCN) have absolute advantages. Existing state-of-the-art (SOTA) methods tended to focus on extracting and identifying features from all bones and joints. However, they ignored many new input features which could be discovered. Moreover, many GCN-based action recognition models did not pay sufficient attention to the extraction of temporal features. In addition, most models had swollen structures due to too many parameters. In order to solve the problems mentioned above, a temporal feature cross-extraction graph convolutional network (TFC-GCN) is proposed, which has a small number of parameters. Firstly, we propose the feature extraction strategy of the relative displacements of joints, which is fitted for the relative displacement between its previous and subsequent frames. Then, TFC-GCN uses a temporal feature cross-extraction block with gated information filtering to excavate high-level representations for human actions. Finally, we propose a stitching spatial–temporal attention (SST-Att) block for different joints to be given different weights so as to obtain favorable results for classification. FLOPs and the number of parameters of TFC-GCN reach 1.90 G and 0.18 M, respectively. The superiority has been verified on three large-scale public datasets, namely NTU RGB + D60, NTU RGB + D120 and UAV-Human.

## 1. Introduction

With the popularity of sensors such as cameras, surveillance cameras and so on, more and more information of human action was documented. How to effectively analyze this information to precisely classify the actions became one of the current hot issues. Skeleton-based methods for action recognition were of great significance in many fields such as intelligent robots, real-time monitoring and human–computer interaction. Human-skeleton-based action recognition methods were usually robust to illumination changes and scene changes, and easy to acquire, where the skeleton-based data were a temporal series of three-dimensional coordinates of the joints. The data could often be obtained through pose estimation methods by using two-dimensional images [1] or directly using sensors such as Kinect cameras [2]. Consequently, many excellent skeleton-based action recognition methods have emerged in recent years [3,4,5,6,7,8,9].

Research on skeleton-based methods could be divided into two stages. In the first stage, the traditional methods based on convolutional neural networks (CNNs) [10,11,12,13,14] and based on recurrent neural networks (RNNs) [4,15,16,17,18] were usually adopted. For CNN-based methods, skeleton data were modeled as virtual images under artificially designed rules. For RNN-based methods, the skeleton data were transformed into a sequence of coordinate vectors representing human joints. In the second stage, the method based on graph convolutional networks gradually became one of the mainstream action recognition methods. GCNs were adept at dealing with non-Euclidean data such as skeleton data, etc. Yan et al. first proposed a spatial temporal graph convolutional network (ST-GCN) for skeleton-based action recognition [5]. Since then, more and more GCN models have been proposed [6,9,19,20,21,22,23].

However, initial GCN-based methods tended to focus on extracting and identifying features from all bones and joints, by using only a small amount of superficial feature information, such as the positions of joints and the lengths of bones, as inputs, which missed many useful input features. Therefore, how to obtain input features with obvious discrimination ability is one of the long-term research hotspots. As for the two-stream adaptive graph convolutional network (2s-AGCN) [8], Shi et al. transformed the first-order information such as the 2D or 3D coordinate information of the joints into joint-stream, and the calculated second-order information such as the lengths and directions of the bones into bone-stream, which were used as feature inputs, then fused and predicted in the end. Such feature species and their fusion methods have achieved excellent results on 2s-AGCN. It was clear that there was still more feature information that could be mined and fed into the model and fused in an early process. PA-ResGCN-N51 [23], SGN [24] and EfficientGCN-B0 [25] innovatively used three input features to fuse early in the model and obtain excellent results in the current GCN-based method.

Today, deep learning models are being developed in real-time network computing as well as on mobile devices. The need for instant computing and deployment on lightweight equipment is imminent. Similarly, this is one of the challenges of GCN-based action recognition methods. Many existing GCN-based models had a large number of parameters, which would lead to a bloated model and an inefficient inference. Therefore, the light weight of models is also a hot direction at present. For example, the quantities of parameters were successively decreased in the following models, which were 6.21 M, 0.77 M, 0.69 M, 0.29 M for RA-GCN [20], PA-ResGCN-N51 [23], SGN [24] and EfficientGCN-B0 [25], respectively. The operation of lightening the model was beneficial for its use in portable devices in the future.

There is a large amount of temporal information to be mined in the time series of skeleton data, which is of great significance for action recognition. It was difficult to extract rich enough feature information in the traditional temporal convolution structure. How to give full play to the role of time information has become one of the factors that researchers pay attention to. Some people began to introduce multi-scale and multi-stream time convolution. Many researchers proposed temporal convolutional blocks with multi-scale structure [26,27,28]. In this paper, a temporal cross-extraction convolution block is proposed in order to obtain high-level temporal information.

Attention mechanisms, as an important part of deep learning, have been studied and modified by many research projects in order to better focus on local information. For example, in PA-ResGCN [23], Song et al. proposed a part-wise attention for assigning different weights to different body parts. In this paper, we propose the SST-Att, which can focus on the local features of basic joints in temporal and spatial dimensions simultaneously. Overall, we have made improvements in four main areas in this paper:

More rich input features are extracted in the preprocess. The relative displacements of the joints between their three frames are used as a type of input features. The relative displacements of two frames, the relative positions of the joints and the geometric information lengths and angles of the bones are used as input features to construct a feature extraction structure with a three-branch configuration during the data preprocessing phase, and the preprocessed features are serially connected immediately after extraction. Results are superior to the method of using speed as an input feature;A temporal feature cross-extraction convolution block is proposed to extract high-level features from temporal information. The cross-extraction convolution block contains a convolution block for cross-extraction of temporal features and a gated CNN unit with information filtering function;The stitching spatial–temporal attention (SST-Att) block is proposed. SST-Att not only considers the spatial attention processing of joints, but also focuses on the remarkable information of joints in the temporal series, so that critical details on these two scales can be further distinguished and extracted;Three large common datasets (NTU RGB + D60, NTU RGB + D120 and UAV-Human) are used to train TFC-GCN in large quantities of experiments, and competitive accuracies are obtained.

The model diagram of TFC-GCN is shown in Figure 1:

## 2. Related Work

*Skeleton-based Action Recognition*. Compared with the traditional RGB-based methods, the methods of skeleton-based action recognition gained increasing notability. In the earlier work, the skeleton-based methods for action recognition mainly relied on CNN and RNN. For example, Ke et al. proposed a representation of skeleton sequences for 3D action recognition based on CNN [11]. Yang et al. proposed an action recognition model based on CNN to make an analysis of action information efficiently [14]. Wang et al. proposed the RNN-based method to learn representations from primitive geometries for skeleton-based action recognition [16]. Zhang et al. proposed an adaptive RNN for high performance from skeleton data. With the development of skeleton-based methods, skeleton data can be well handled by GCN-based methods. Yan et al. firstly proposed spatial temporal graph convolutional network (ST-GCN) for skeleton-based action recognition to model in the temporal dimension and spatial dimension simultaneously [5]. Shi et al. proposed a two-stream adaptive graph convolutional network (2s-AGCN) so as to improve the accuracy of model significantly [8]. Song et al. proposed a richly activated graph convolutional network (RA-GCN) to explore richer features from more vibrant joints and solve the problem of the human body being occluded [20]. Hou et al. proposed a GCN-based effective multi-channel mechanism for extracting local and global action information, which could improve the performance of GCN-models [22]. Chen et al. proposed a channel-wise topology graph convolutional network (CTR-GCN) to learn diverse structures of topology and aggregate the features of joints effectively [21]. The skeleton-based GCN method has greatly promoted the development of action recognition and has also become one of the mainstream action recognition methods.

*The inputs of models.* The input features have always been among the important issues in the field of behavior recognition. Most GCN-based action recognition methods had only one or two input features and fused multiple features later in training. The 2s-AGCN [8] divided the input of skeleton data into two main streams (bone-stream and joint-stream) and fused them in the later stage of training, which was an effective method of data augmentation. Later, ResGCN [23], with a third-stream branch, was proposed by Song et al., where feature extraction and fusion were carried out in the early stage of the training. This operation could reduce parameter redundancy and the complexity of the model. The input and fusion of features play a crucial role in the model training process. The input of features can determine the model’s ability to distinguish samples early, and the model’s fusion period can determine the training and evaluating steps of the model. Discriminating input features improve accuracy, and early feature fusion reduces parameter redundancy and improves model performance.

*Temporal feature extraction.* Some methods of skeleton-based GCN for action recognition did not pay sufficient attention to the information in the temporal dimension although they showed great advantages in the processing of non-Euclidean data. Recently, many endeavors are focused on further extracting information in the temporal dimension. Kong et al. created a multi-scale temporal transformer block in his proposed MTT-GCN [28] to extract the temporal features of high-order dimensions and obtain an easily distinguishable temporal information representation, which achieved excellent performance in the subsequent classification process. Chen et al. added a multi-scale temporal convolution block to his CTR-GCN [21], extracted the temporal features in different streams and obtained more differentiated temporal features.

*Attention mechanism.* In the development of neural computing networks, attention mechanisms have become one of the essential model components. The spatial–temporal attention based long short-term memory (LSTM) [4] was applied to skeleton-based action recognition, the role of assigning different levels of attention to different joints with different weights in each frame. Chan et al. proposed the squeeze-and-excitation network as the attention block of spatial and temporal channels in their GAS-GCN, which gave high weight to important channels [29]. Song et al. proposed a part-wise attention mechanism in their PA-ResGCN to discover the different importance of different body parts throughout the action sequence and to give higher weights to more important body parts [23].

*Lightweight models.* The huge amount of computation and excess of parameters often limited the operation of the model on small devices, so it was one of the hot spots to lighten the model for better efficiency. The ResGCN-N51 [23] added a bottleneck block to cut the parameters of the model, which was only 0.77 M. The Ghost-GCN [30] proposed by Jang et al. had a fairly low number of parameters, just 0.51 M. Later Song et al. optimized and reconstructed the original ResGCN model, and the obtained EfficientGCN [25] with only 0.29 M parameters. The above-mentioned methods achieved low parameter quantities while maintaining high accuracy through different models. In our opinion, with the urgent need for lightweight models in mobile devices, the number of parameters of the model will decrease in future research. In subsequent experiments, we will study how to reduce the number of parameters of the model to improve the efficiency of the models.

## 3. Methodology

In this section, we will discuss the details of our proposed TFC-GCN for skeleton-based model in action recognition. Firstly, we introduce the three-branch input in TFC-GCN and highlight our proposed feature extraction strategy of the relative displacement between three adjacent frames. This innovative input feature can give a new classification basis to the model extraction process. In the process of classifying each sample, relative displacement can obtain better results than traditional velocity of joints. It contains relative displacement information that not only represents the displacement of the joint within three frames, but also implicitly contains velocity information between three frames, so the information in the included bone sequence is richer, more effective and discriminating. Secondly, we introduce a temporal feature cross-extraction convolution block that can dig out rich temporal information. Compared with the traditional temporal information extraction process, the temporal cross-extraction convolution superimposes the feature channels in the process of continuously discovering deep features and passes the processed data to the next process of cross-extraction convolution. At the same time, we carry out a multi-stream parallel structure for the process of time cross extraction convolution, so that the features extracted by the temporal cross-extraction convolution block are more detailed and more stable. Finally, the SST-Att to improve feature discrimination will be demonstrated. SST-ATT divides the temporal features and spatial features and extracts the weights at the same time, then strengthens the high-weight part and weakens the low-weight part and finally combines the two to obtain the final weighted tensor. This method can better explore important parts of the human skeleton sequence and provide effective effects for the final classification. The whole process of TFC-GCN preprocessing for raw skeleton data and feature extraction is shown in Figure 2:

### 3.1. Graph Convolutional Network

GCN could play a significant role in processing data of graph structures such as the skeleton data. Considering the joint points as the vertices of the graph, the connections of skeletons between the points could be regarded as the edges of the graph. The skeleton data are defined as graph G=(V,E), where V denotes the set of M vertices, for which V={vi,i=1,2,3,…,M}. E is the set of N edges, for which E={ej, j=1,2,3,…,N}, where ej is the connection of two joint vertices. Then the convolution operation can be formulated as Equation (1):(1)fout=σ(Λ−12(A+IN)Λ−12finM)
where fin∈ℝN×Cin×T and fout∈ℝN×Cout×T denote the input and output features, respectively, and N, Cin/out, T denote the numbers of vertices, channels and frames, respectively. A denotes the adjacency matrix, which is used to aggregate the information of adjacent nodes. A+IN denotes the adjacency matrix plus the self-connected unit matrix  IN of the graph. Λ denotes the degree matrix, which is used for the normalization of A+IN. M denotes a learnable weight matrix. σ(·) denotes the activation function.

In order to realize the convolution of joint points in the spatial dimension, we convert Equation (1) into the formula as shown in Equation (2):(2)fout=σ(∑kKsW(Avfin)⊙Mv)
where W∈ℝCin×Cout×1×1 denotes the weight vector of 1 × 1 convolution. Mv∈ℝN×N denotes the attention matrix, which shows the importance of each vertex in the graph. Av∈ℝN×N denotes the normalized adjacency matrix, which equals Λ−12(A+IN)Λ−12. Finally, Ks denotes the kernel size and ⊙ denotes the dot product.

### 3.2. Preprocessing

Preprocessing of raw data is important throughout the training process. The existing skeleton-based GCN methods mainly preprocessed skeleton data into three types of features which can be obtained by different sensors: (1) joint position, (2) movement speed and (3) bone geometric features. In our model, we retain the two distinct features of the relative position of the joint and the geometric features of the bones and use the relative displacement of all joints to the joints in the center of the adjacent frame instead of the speed of movement and achieve good results. The calculation diagram of the three input features is shown in Figure 3:

In the determined 3D coordinate system, the input dimension of the action sequence is x∈ℝCin×Tin×Vin, where Cin, Tin and Vin represent input coordinates, sequences of frames and joints, respectively.

Relative position set is represented as $P=\{p_{i}|i=1,2,\ldots ,V_{in}\}$ and the $p_{i}$ (is) calculated by formulas such as Equation (3):(3)$$p_{i}=x\left[:,:,i\right]-x\left[:,:,s\right]$$
where s represents the central spine joint. The resulting $p_{i}$ represents the relative position of the individual joints relative to the central spine joint. We use the set $D=\{d_{t}|t=0,1,2,\ldots ,T_{in}\}$ to represent the relative displacement. The relative displacement *d_t_*, is calculated by Equation (4):(4)$$d_{t}=x\left[:,t,i\right]=x\left[:,t-1,i\right]-x\left[:,t+1,s\right]$$
where s represents the central spine joint and i represents the ith joint and t represents a certain frame in which the current raw skeleton data are located and the resulting dt represents the relative displacement between frame t−1 and frame t+1 recorded on frame t. Since the temporal length between each frame is concurrent, the speed information is implicitly contained in $d_{t}$ simultaneously. The geometric features of the bones are contained in the sets $L=\{l_{i}|i=1,2,\ldots ,V_{in}\}$, $A=\{a_{i}|i=1,2,\ldots ,V_{in}\}$ and $S=\{s_{i}|i=1,2,\ldots ,V_{in}\}$, where $l_{i}$ represents the bone length, $a_{i}$ represents the angle of the bones, $s_{i}$ represents the ratio of the difference in bone movement angle to the relative velocity between the two endpoints of the bone over a certain temporal length and the three feature data are formulated as Equation (5):(5)$$\{\begin{array}{l} l_{i}=x\left[:,:,i\right]-x\left[:,:,i_{adjacent}\right]\\ a_{i}=\arccos\left(\frac{l_{i,joint}}{\sqrt{l_{i,x}{}^{2}+l_{i,y}{}^{2}+l_{i,z}{}^{2}}}\right)\\ s_{i}=\frac{a_{i}-a_{j}}{x\left[:,t+2,i\right]-x\left[:,t,j\right]} \end{array}$$
where the $i_{adjacent}$ represents the adjacent joints for the joint i. joint∈{x,y,z} represents the joint in the 3D coordinate system.

### 3.3. Temporal Feature Cross-Extraction Convolution

The temporal series of human movements are rich in information, so modeling in the temporal dimension should receive vital attention. We propose a temporal convolution block that can cross-extract temporal features with a gated information filtering function. In the process of cross-extraction, we obtain high-level features in the temporal dimension by feature extractions, and stitch them with the secondary originally retained features to obtain more distinguishable feature data. At the same time, the gating unit can obtain new features for efficient and rapid filtering and extraction of information, retaining useful information related to classification and discarding useless information. Gated CNN was first proposed by Dauphin et al. for natural language processing [29]. Compared to LSTM, Gated CNN is not only more computationally efficient, but also more effective, which makes it excellent to be used to process bones in temporal series. The flowchart of temporal feature cross-extraction convolution block is shown in Figure 4.

For Figure 4, the temporal convolution block contains four-stream convolution methods, which are divided into (1) point convolution; (2) point convolution with pooling; (3) temporal feature cross-extraction convolution and (4) gated CNN unit. Through four-stream temporal convolution, we can obtain richer features. We will continue to analyze the convolution process in subsequent articles. The structural details of the convolutional blocks involved in the convolution process are shown in Figure 5:

The point convolution and full convolution blocks are formulated as Equation (6):(6)$$\{\begin{array}{l} Point\_Conv1\left(X\right)=BN(X_{C_{in}}*W_{1}+b_{1}{)}_{C_{inner}}\\ Point\_Conv2\left(X\right)=BN(X_{C_{inner}}*W_{2}+b_{2}{)}_{C_{in}}\\ Point\_Conv3\left(X\right)=ReLU(BN(X_{C_{in}}*W_{3}+b_{3}{)}_{C_{inner}})\\ Full\_Conv1\left(X\right)=BN(X_{C_{in}}*W_{4}+b_{4}{)}_{C_{in}}\\ Full\_Conv2\left(X\right)=BN(X_{C_{in}}*W_{5}+b_{5}{)}_{C_{inner}} \end{array}$$where X is the input feature. $W_{i}\;\left(i=1,2,3,4,5\right)$ are the convolution kernels of different convolution blocks. $b_{i}\;\left(i=1,2,3,4,5\right)$ are the bias terms that can be trained. BN(·) and ReLU(·) represent BatchNorm2d operations and ReLU activation function, respectively. The subscript $C_{in}$ indicates the number of input channels and $C_{inner}$ indicates the number of output channels at operation. We will introduce their principles and functions separately:

Point Convolution (PC). This operation can convert the input features into a more high-leveled representation, and at the same time ensure a little amount of original high-level information to prevent large-scale information loss. Point convolution process is expressed as Equation (7):(7)$$P_{out}=ReLU\left(BN(H*W+b)\right)$$
where H is the input feature. W is the convolution kernel. b is a bias term that can be trained. BN(·) represents BatchNorm2d normalization operation. ReLU(·) is the ReLU activation function. $P_{out}$ is the output of point convolution.

Point convolution performs feature extraction on a quarter of the input feature channel. On the one hand, we can obtain high-level features under a single convolution with a small amount of computing resources, and on the other hand, this operation can prevent excessive feature extraction of some samples and cause feature confusion, which greatly ensures the stability of output features.

Point Convolution with Pooling (PCP). This operation adds pooling operations to the point convolution and passes the input feature map through two parallel MaxPool2d layers and AvgPool2d layers. More comprehensive and richer high-level features can be extracted by the parallel dual-pooling layer, and at the same time, redundant information can be removed, insignificant features can be compressed and the sensing field can be expanded, which can reduce the quantity of parameters and retain helpful information. The process of point convolution with pooling is expressed as Equation (8):(8)$$P_{p\_out}=AvgPool\left(H*W_{1}+b_{1}\right)⊕MaxPool\left(H*W_{2}+b_{2}\right)$$
where H represents the input feature. $W_{1}$ and $W_{2}$ are two temporal convolution kernels with the same size. $b_{1}$ and $b_{2}$ are two bias terms that can be trained. ⊕ is the addition operation. AvgPool(·) and MaxPool(·) represent the average pooling operation and the maximum pooling operation, respectively.

Compared with the PC-stream, the pooling operation is added to the PCP-stream to obtain a differentiated feature stream while saving the computing resources of the feature.

Cross-extraction (CE). This operation uses multiple point convolution blocks for cross-convolution, and two full convolution blocks are used at the beginning and end of training, respectively. In the process of convolution, the output features of the previous convolution block are directly connected with the input features of the current round, the significant features in the data are continuously strengthened and extracted and the two Swish activation functions are used to non-linearly activate the feature tensor, discarding useless information and retaining the information that plays a key role in classification. The operation of cross-extraction convolution is expressed as Equation (9):(9)$$\{\begin{array}{l} H_{1}=Swish\left(Full\_Conv1\left(H\right)\right)\\ H_{1}^{\prime }=Point\_Conv1\left(H_{1}\right)\\ H_{2}=Point\_Conv1\left(H⊕H_{1}\right)\\ H_{3}=Swish\left(Point\_Conv2\left(H_{2}\right)\right)\\ H_{3}^{\prime }=Swish\left(Point\_Conv2\left(H_{1}^{\prime }⊕H_{2}\right)\right)\\ C_{out}=Full\_Conv2\left(H_{3}⊕H_{3}^{\prime }\right) \end{array}$$
where H is the input feature of the previous layer. $H_{1},\;H_{1}^{\prime },\;H_{2},\;H_{3},\;H_{3}^{\prime }$ represent the transfer features produced in the intermediate process. $C_{out}$ represents the final output value of the process of cross-extraction convolution. ⊕ is the addition operation.

Gated CNN Unit (Gate). Gated CNN is able to model based on contextual information in the time dimension. The gating mechanism is generated by Sigmoid. The output feature is another first-class convolution output multiplied by the gated feature output. At the same time, the operation we add before the Sigmoid function can enhance the channel weight of the information of the gating process and enhance the function of gating, so as to retain the information that affects the recognition result and discard the useless information. The Gated CNN process can be described with Equation (10):(10)$$\{\begin{array}{l} Channel\left(X\right)=Sigmoid\left(Linear\left(ReLU\left(Linear\left(AvgPool\left(X\right)\right)\right)\right)\right)\\ G_{1}=H*W_{1}+b_{1}\\ G_{2}=Channel\left(H*W_{2}+b_{2}\right)\\ G_{output}=G_{1}\;⊗\;Sigmoid(G_{2}) \end{array}$$
where Channel(·) represents the channel enhancing process, AvgPool(·), Linear(·), ReLU(·) and Sigmoid(·) represent average pooling operations, linear transformation operations, ReLU(·) activation functions and Sigmoid(·) activation functions, respectively. The specific details are shown in Figure 5. H represents the input features. $W_{1}$ and $W_{2}$ are two temporal convolution kernels of the same size. $b_{1}$ and $b_{2}$ are two bias terms that can be trained. Sigmoid(·) is the sigmoid function. ⊗ stands for the element-wise product. $G_{output}$ represents the final output of the Gated CNN.

The specific details of temporal feature cross-extraction convolution are shown in Figure 6:

### 3.4. The Stitching Spatial–Temporal Attention Mechanism

In the past, the attention mechanism of skeleton-based action recognition models was mainly implemented by multi-layer perception. The parts of the body were carried out in different channels and spatial–temporal dimensions, and the other feature dimensions were treated as a single unit for global meaning. Taking Pa-ResGCN as an example, its attention mechanism Pa-Att focuses on the importance of different body parts for the entire action sequence, performs feature extraction on different channels and obtains different importance levels for different parts, but this is limited to attention in the spatial dimension. In fact, there is also a certain correlation between the critical features in the temporal dimension besides the spatial dimension. The proposed SST-Att can pay attention to the relationship between the temporal dimension and the spatial dimension, so that the important joints in those important timeframes are valued. The specific implementation flowchart of SST-Att is shown in Figure 7:

We express the process of SST-Att as Equation (11):(11){FCN(X)=HardSwish(BN(X∗W+b))Xinner=FCN(AvgPoolT(Xin)⊕AvgPoolV(Xin))Xout=Xin⊙((Xinner∗WT+bT)⊗(Xinner∗WV+bV))
where X, Xin and Xout are the input features of FCN(·), the input features of SST-Att and the output features, respectively. HardSwish(·) stands for the HardSwish activation function. BN(·) stands for the BatchNorm2d normalization operation. AvgPoolT(·)  and AvgPoolV(·) represent the average pooling operation on channels of T and V, respectively. ⊕, ⊗ and ⊙ represent concatenation operations, tensor multiplication operations and element dot multiplication operations, respectively. W∈ℝC×Cr, WT∈ℝCr×C and WV∈ℝCr×C are the weight parameters of the convolution process, and b, bT and bV are trainable bias terms.

### 3.5. Evaluation for the Light weight of Model

In order to evaluate the light weight of the model, we propose a parameter and its calculation method. In other words, what we are looking forward to is that the model maintains a high evaluating accuracy with a low quantity of parameters. Expressing the measurement parameters as Equation (12):(12)$$\theta_{lw}=ln\left(\frac{\alpha_{acc}}{\rho_{param}}+1\right)$$$\theta_{lw}$ represents our measure called ‘Lightweight Value’. $\alpha_{acc}$ represents the final evaluating accuracy of the model. $\rho_{param}$ represents the quantity of parameters of the model. To achieve the ideal model, $\alpha_{acc}$ needs to be as large as possible and $\rho_{param}$ as small as possible. Choosing this calculation method can well achieve (meet) our measurement needs and compress the number of $\theta_{lw}$. Of course, under our scale, we should strive to obtain the maximum value of $\theta_{lw}$.

## 4. Experimental Results

In this section, the results of TFC-GCN on three large public datasets (NTU RGB + D60, NTU RGB + D120 and UAV-Human) are presented. The contribution of each improvement is verified in the following ablation experiments and extensive comparisons with the SOTA methods.

### 4.1. Datasets

NTU RGB + D60. This dataset [31] consisted of 60 action classifications taken from 56,680 human action videos collected from three Kinect v2 cameras indoors. Each video consisted of 300 frames, of which less than 300 frames are padding with zeros, and contained one or two skeletons, each consisting of 25 joints. There are two recommended benchmarks for this dataset: (1) Cross-view (X-sub). The benchmark consisted of 40 subjects divided into two groups, containing 40,320 training sample videos and 16,560 validation sample videos. (2) Cross-view (X-view). This benchmark consisted of 37,920 training sample videos collected by cameras 2 and 3 and 18,960 validation sample videos collected by camera 1.

NTU RGB + D120. This dataset [32] was an extended version of the NTU RGB + D60 dataset (from 60 classes to 120 classes), containing 114,480 videos from 155 viewpoints with 106 participants. There were also two recommended benchmarks for this dataset: (1) cross-subject (X-sub120). In this benchmark subjects are divided into two groups and a training set containing 63,026 videos and a validation set containing 50,922 videos were constructed. (2) cross-subject(X-set120). This benchmark included 54,471 training video samples and 59,477 validation video samples, which were divided according to the horizontal and vertical positions of the camera relative to the subjects.

UAV-Human. This dataset [33] was collected by a flying drone in multiple urban and rural areas during the day and night over 3 months, so it covered a wide variety of subjects, backgrounds, lighting, weather, occlusion, camera movements and drone flying attitudes. For the understanding and analysis of human behavior in videos acquired by UAV, UAV-Human contained 67,428 multi-mode video sequences and 119 targets for action recognition, 22,476 frames for posture classification, 41,290 frames and 1144 frames for gender reidentification and 22,263 frames for attribute identification. Such a comprehensive and challenging benchmark will be able to facilitate UAV-based research into human behavior understanding, including motion recognition, posture estimation, re-recognition and attribute recognition.

### 4.2. The Details of Implementation

In our experiments, the initial learning rate is set to 0.1 with the end epoch of 70, which decays to 10 at the 50th epoch. In the first 10 rounds, a warm-up strategy was used to gradually increase the learning rate by 0.01 per epoch and from 0.1. The stochastic gradient descent (SGD) with Nesterov momentum of 0.9 was used and the attenuation weight of 0.0001 was used to optimize the training parameters. All trials were conducted on one RTX 4090 GPU.

### 4.3. Comparisons with SOTA Methods

#### 4.3.1. NTU RGB + D60 and NTU RGB + D120

We performed experiments on four benchmarks of the NTU RGB + D60 and NTU RGB + D120 datasets and compared them with the SOTA methods. The results are shown in Table 1:

It is clear that TFC-GCN experimental results on four benchmarks are in a weak position compared to other SOTA methods. However, to our best knowledge, TFC-GCN has only 0.18 M parameters, which is the model with the smallest number of parameters so far and has only 62% those of the EfficientGCN-B0 in Table 1.

#### 4.3.2. UAV-Human

We performed experiments on two benchmarks of the UAV-Human dataset and compared them with the SOTA methods. The results are shown in Table 2:

To express the idea that lightweight models are easy to deploy, we conducted experiments on two benchmarks of the UAV-Human dataset. As can be seen from Table 2, TFC-GCN has a significant advantage over the benchmark CSv1 and has the lowest number of parameters, while it has considerable room for improvement on CSv2.

#### 4.3.3. Performance of Model

Among the four properties in Table 3, TFC-GCN has considerable advantages over other SOTA methods in terms of parameter quantity, FLOPs and lightweight value $\theta_{lw}$ TFC-GCN ensures high evaluating accuracy while having a very low number of parameters, which meets our “lightweight” assumption.

### 4.4. Ablation Experiment

In this section, we perform ablation experiments on TFC-GCN for our proposed improvements, the results of which will be presented below.

#### 4.4.1. Input Features

To compare the effectiveness of different input features in the model and verify the advantages of relative displacement, we perform an ablation experiment on TFC-GCN. J, V, B and C represent joint position, speed, bone geometry and relative displacement of joints, respectively. The experimental results are shown in Table 4:

As can be seen from Table 4, the relative displacement has a significant advantage over the other three input characteristics in a single-stream input. In the three-stream input, better results are also achieved with JBC as input.

#### 4.4.2. Temporal Feature Cross-Extraction Convolution

In order to verify the properties of our proposed temporal feature cross-extraction convolution block in the combination, we performed another ablation experiment on multiple convolutional block combinations on the basis of CE block. CE, PC, PCP and Gate represent point convolution, point convolution with pooling, cross-extraction convolution and gated CNN unit in Section 3.2, respectively.

As can be seen from Table 5, the temporal convolution block with four-stream input has the highest evaluating accuracy, and the number of parameters is only 0.1 M higher than that of the CE block with single-stream input.

### 4.5. Visualization of Results

We visualized the TFC-GCN on an X-Sub benchmark. The confusion matrix and heat map are shown in Figure 8 and Figure 9, respectively:

The confusion matrix is a kind of indicator to judge the results of the model, the diagonal line of the confusion matrix indicates the number of samples correctly predicted, and the denser the predicted values are distributed diagonally, the better the performance of the performance of the model. Unlike Figure 8, the confusion matrix can not only visually represent the classification accuracy, but also give the probability of classification and misclassification. As shown in Figure 8, ‘jump up’ achieves 100% classification accuracy, but ‘writing’ only achieves 58% classification accuracy. As can be seen from the figure, another action that is easy to cause the misclassification of ‘writing’ is ‘typing on a keyboard’. Therefore, we speculate that one of the reasons for the low probability of correct classification of ‘writing’ is that this action uses a lot of hands, which is easy to misclassify with other hand movements in the classification process. The confusion matrix in Figure 8 visually represents the classification results of TFC-GCN. It is obvious that the results of the classification are almost straight in the confusion matrix, which is a good representation of the classification results.

In order to show the division of important joints in the human action sequence by the attention mechanism, we generated a joint point heat map of four movements. The results are shown in Figure 9:

In the heat map, we use red to represent important joints in the action sequence, and the more vivid the red, the greater the role of the joint in this action. In Figure 9, we show the heat concentration of the four actions ‘cheer up’, ‘pointing to something with finger’, ‘sitting down’ and ‘wear on glasses’ on each of the 10 frames, and TFC-GCN can pay good attention to the important joints in those action sequences. The heat map displayed is a visual representation of the attention paid by TFC-GCN to important joints in the human sequence. In our opinion, TFC-GCN played an ideal role in the movements shown, focusing on important joints and thus having a positive effect on classification.

## 5. Conclusions

In this paper, we propose a high-efficiency model TFC-GCN with low parameter quantity for action recognition. The relative displacement proposed as an input feature can better fit the joint position information between the front and back frames, and multi-branching early fusion with other inputs can also reduce parameter redundancy. Our proposed temporal feature cross-extraction proposal strengthens the attention to time series in action recognition and can further extract rich temporal information. In addition, the proposed stitching spatial–temporal attention block can also play a significant role in recognizing important spatiotemporal information in action sequences. We conduct experiments on three datasets NTU RGB + D60, NTU RGB + D120 and UAV-Human and obtain good evaluating accuracy, while the number of parameters and FLOPs are much smaller than other SOTA models, and the lightweight value ‘A’ is significantly higher than other SOTA models. With the trend of the rapid development of Edge Computing technology in IoT, there is an urgent need for deployment on mobile and embedded devices, so we believe that the proposed model has considerable potential.

## Figures and Tables

**Figure 1 sensors-23-05593-f001:**
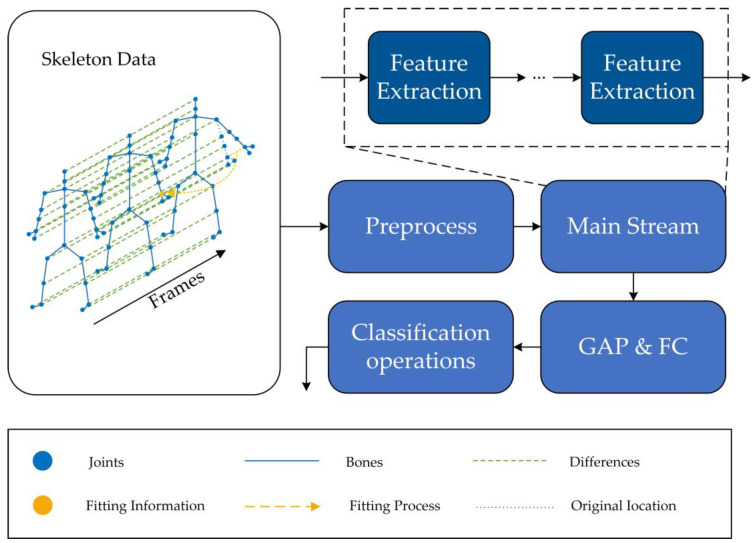
The model diagram of TFC-GCN, where GAP and FC represent the global average pooling layer and the fully connected layer, respectively.

**Figure 2 sensors-23-05593-f002:**
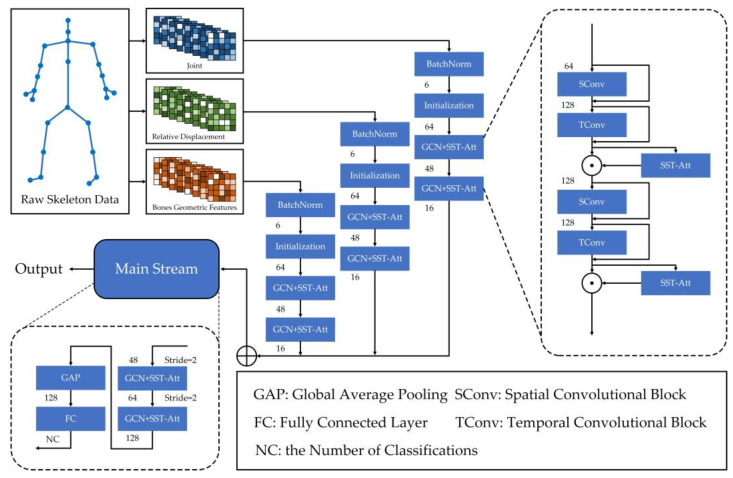
The whole process of TFC-GCN, where ⊕ indicates that the streams are concatenated in series, and ⊙ indicates the product of the output data and the attention weight obtained from SST-Att. The number between the blocks at each step represents the number of the feature channels. The raw skeleton data are first processed into three streams of input features, which are initialized and extracted respectively. After that, the three features are fused and fed into the mainstream network for feature extraction. Finally, the classification is carried out.

**Figure 3 sensors-23-05593-f003:**
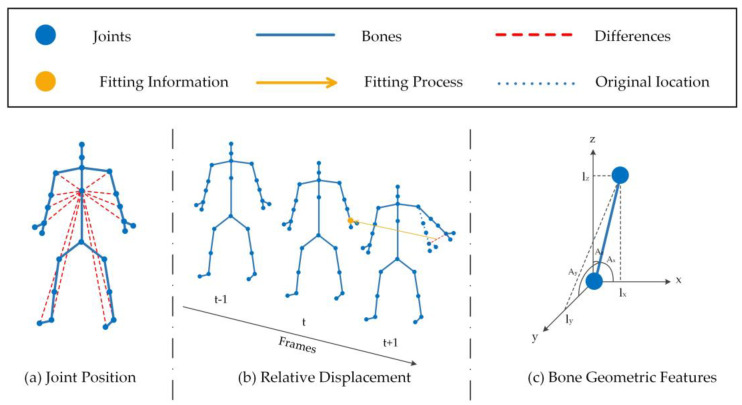
Illustration of calculation of the input skeleton data. (**a**) Joint position, (**b**) Relative displacement, (**c**) Bone geometric features, where (**a**) represents the position of each joint relative to the central spine joint. (**b**) represents the relative displacement of each joint point from frame t−1 to frame t+1 corresponding to the same joint point at frame t. (**c**) represents the three-dimensional joint coordinate $\;\left(l_{x},l_{y},l_{z}\right)$ and the three-dimensional angle $\left(A_{x},A_{y},A_{z}\right)$ of the bone in a 3D coordinate system.

**Figure 4 sensors-23-05593-f004:**
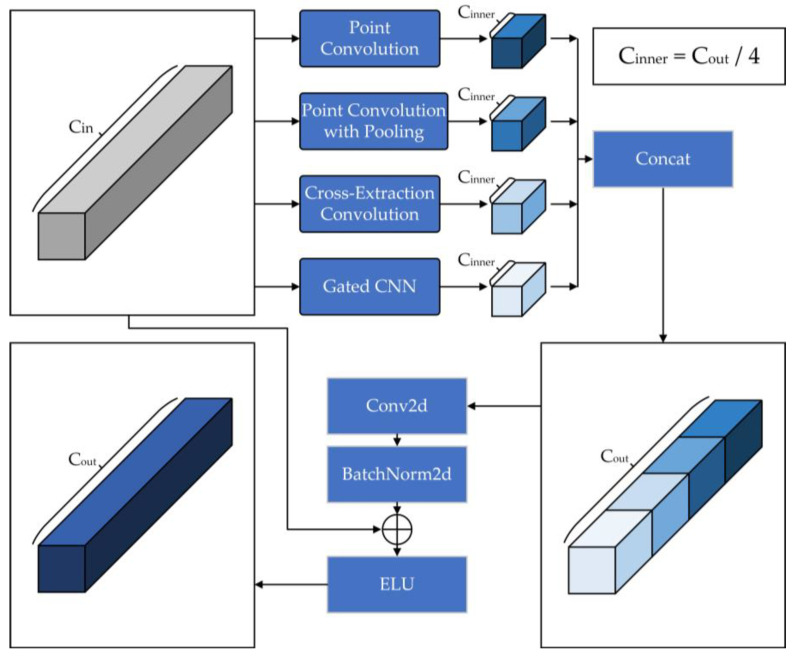
The four-stream structure of the input data, where ⊕ represents the sum of tensors. Each stream has a different degree of feature extraction.

**Figure 5 sensors-23-05593-f005:**
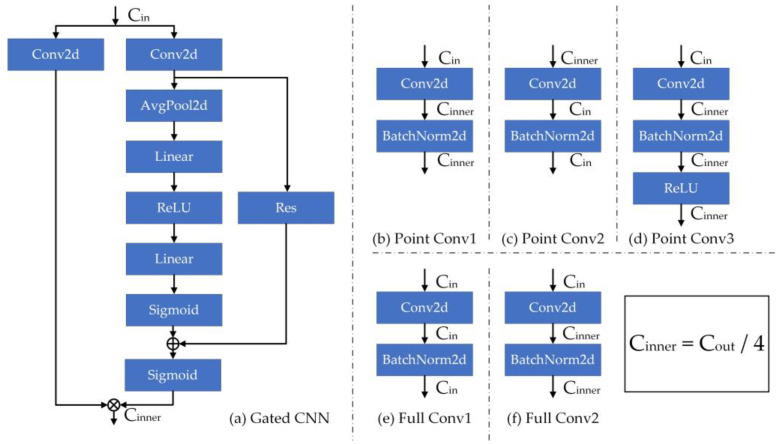
Implementation details of the point convolution, full convolution and gated convolution, where ⊕ and ⊗ represent the sum of tensors and the dot multiplication of tensors, respectively.

**Figure 6 sensors-23-05593-f006:**
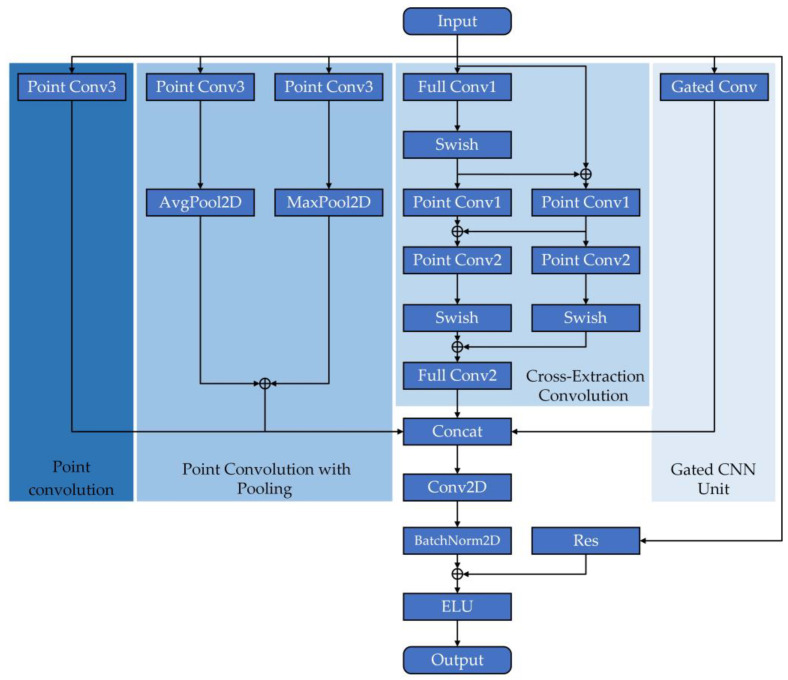
The specific details of temporal feature cross-extraction convolution, where ⊕ represents the sum of tensors.

**Figure 7 sensors-23-05593-f007:**
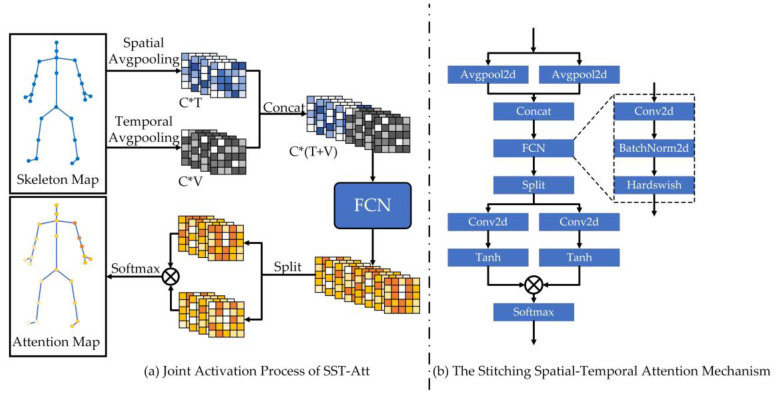
The specific implementation flowchart of SST-Att. (**a**) Joint activation process of SST-Att. (**b**) The stitching spatial–temporal attention mechanism, where ⊗ represents tensor multiplication.

**Figure 8 sensors-23-05593-f008:**
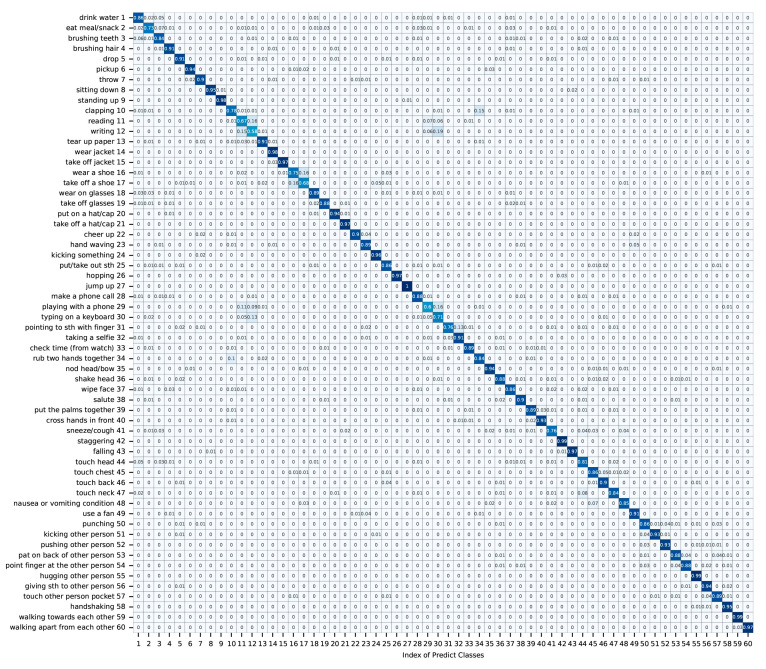
Confusion matrix. The shade of the blue patch represents the numeric value of the evaluating accuracy. The higher the evaluating accuracy, the darker the blue.

**Figure 9 sensors-23-05593-f009:**
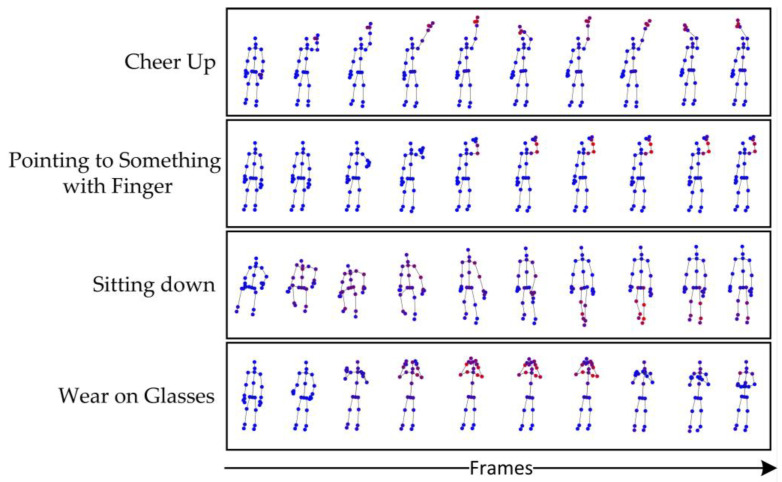
Heat map. The red nodes represent the activated joints, and the darker the red, the more important the joints. Blue nodes represent nodes that are not activated.

**Table 1 sensors-23-05593-t001:** Experimental results on NTU RGB + D60 and NTU RGB + D120 and comparison with other SOTA methods. In Table 1, the X-Sub, X-View, X-Sub120, X-Set120 column headings represent the evaluating accuracy of four benchmarks.

Model	Param. (M)	Acc.			
		X-Sub (%)	X-View (%)	X-Sub120 (%)	X-Set120 (%)
ST-GCN [5]	3.10 *	81.5	88.3	70.7 *	73.2 *
SR-TSL [34]	19.07 *	84.8	92.4	74.1 *	79.9 *
RA-GCN [20]	6.21 *	87.3	93.6	81.1	82.7
AS-GCN [19]	9.05 *	86.8	94.2	77.9 *	78.5 *
2s-AGCN [8]	6.94 *	88.5	95.1	82.5 *	84.2 *
AGC-LSTM [7]	22.89	89.2	95.0	—	—
DGNN [9]	26.24	89.9	96.1	—	—
SGN [24]	0.69	89.0	94.5	79.2	81.5
Ghost-GCN [30]	0.51	89.0	94.6	—	—
PL-GCN [35]	20.70	89.2	95.0	—	—
NAS-GCN [36]	6.57	89.4	95.7	—	—
ResGCN-N51 [23]	0.77	89.1	93.5	84.0	84.2
JOLO-GCN [37]	10.42	93.8	98.1	87.6	89.7
CTR-GCN [21]	1.46	92.4	96.8	88.9	90.6
Dynamic-GCN [38]	14.40	91.5	96.0	87.3	88.6
MST-GCN [39]	12.00	91.5	96.6	87.5	88.8
EfficientGCN-B0 [25]	0.29	90.2	94.9	86.6	85.0
DeepActsNet [40]	19.4	94.0	97.4	89.3	88.4
SGN-SHA [41]	0.32	87.5	92.6	78.5	87.1
CGCN [42]	—	92.9	96.0	—	—
ST-AGCN [43]	—	88.2	94.3	—	—
TFC-GCN (ours)	0.18	87.9	91.5	83.0	81.6

*: These results are provided by EfficientGCN [25] on the released codes by Song et al.

**Table 2 sensors-23-05593-t002:** Experimental results on UAV-Human and compare with other SOTA methods. We calculated A on the CSv1 benchmark.

Model	Param. (M)	CSv1 (%)	CSv2 (%)	$\theta_{lw}$
DGNN [9]	26.24	29.9	—	0.76
ST-GCN [5]	3.10	30.3	56.1	2.38
2s-AGCN [8]	6.94	34.5	66.7	1.79
CTR-GCN [21]	1.46	41.7	—	3.79
EfficientGCN-B0 [25]	0.29	39.2	63.2	4.91
Shift-GCN [44]	—	38.0	67.0	—
MKE-GCN [45]	1.46	43.8	—	3.43
STGPCN [46]	1.70	41.5	67.8	3.24
HARD-Net [47]	—	36.97	—	—
TFC-GCN (ours)	0.18	39.6	64.7	5.40

**Table 3 sensors-23-05593-t003:** Experimental results on X-Sub to compare performance of models with other SOTA methods.

Model	Param. (M)	Acc. (%)	FLOPs (G)	$\theta_{lw}$
ST-GCN [5]	3.10 *	81.5	16.32 *	3.31
SR-TSL [34]	19.07 *	84.8	4.20 *	1.70
RA-GCN [20]	6.21 *	87.3	32.80 *	2.71
AS-GCN [19]	9.50 *	86.8	26.76 *	2.32
2s-AGCN [8]	6.94 *	88.5	37.32 *	2.62
SGN [24]	0.69	89.0	—	4.86
Ghost-GCN [30]	0.51	89.0	—	5.16
PL-GCN [35]	20.70	89.2	—	1.67
NAS-GCN [36]	6.57	89.4	—	2.68
ResGCN-N51 [23]	0.77	89.1	—	4.76
JOLO-GCN [37]	10.42	93.8	41.80	2.30
CTR-GCN [21]	1.46	92.4	—	4.16
Dynamic-GCN [38]	14.40	91.5	—	2.00
MST-GCN [39]	12.00	91.5	—	2.15
EfficientGCN-B0 [25]	0.29	90.2	2.73	5.74
DeepActsNet [40]	19.4	94.0	6.6	1.76
TFC-GCN (ours)	0.18	87.9	1.90	6.20

*: These results are provided by EfficientGCN [25] on the released codes by Song et al.

**Table 4 sensors-23-05593-t004:** Ablation experiment of input features.

Input	Param. (M)	Acc. (%)
J	0.12	85.1
V	0.12	83.4
B	0.12	85.0
C	0.12	86.3
JVB	0.18	87.6
JVC	0.18	86.9
VBC	0.18	87.3
JBC	0.18	87.9

**Table 5 sensors-23-05593-t005:** Ablation experiment of temporal convolution block.

Temporal Convolution	Param. (M)	Acc. (%)
Only CE	0.17	86.7
CE + PC	0.17	87.2
CE + PCP	0.17	87.2
CE + Gate	0.17	87.2
CE + PC + PCP	0.18	87.3
CE + PC + Gate	0.18	87.5
CE + PCP + Gate	0.18	87.0
CE + PC + PCP + Gate	0.18	87.9

## Data Availability

We will provide links to the three datasets used in our experiments. NTU RGB+D60 and NTU RGB+D120: https://rose1.ntu.edu.sg/dataset/actionRecognition/. UAV-Human: https://sutdcv.github.io/uav-human-web/.

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
