# Peer review of "TFC-GCN: Lightweight Temporal Feature Cross-Extraction Graph Convolutional Network for Skeleton-Based Action Recognition"

_sensors, 2023, doi:10.3390/s23125593_

Round 1

Reviewer 1 Report

The methodology retains two distinct features, the relative position of the joints and the geometric features of the bones, for preprocessing the skeleton data. It uses the relative displacement of joints instead of movement speed, which has shown promising results. By considering joint positions, relative displacements, and bone geometric features, the model can capture important characteristics of the human movements. The methodology introduces several components, such as GCN, feature extraction strategy, and temporal feature cross-extraction convolution block. This complexity might increase the computational requirements and training time of the model. It could be challenging to implement and train the model, especially without detailed implementation guidelines. The description of the methodology is brief and lacks in-depth explanations or justifications for design choices. More detailed explanations, including the underlying rationale and theoretical foundations, would enhance the understanding of the proposed approach. Lightweight model: TFC-GCN has only 0.18 million parameters, which is the smallest among the compared methods. This makes it computationally efficient and easier to deploy. Despite being a lightweight model, TFC-GCN achieves competitive accuracy on the benchmark datasets. It achieves an accuracy of 87.9% on X-Sub, demonstrating its effectiveness in human action recognition tasks. The ablation experiment shows that the inclusion of relative displacement as an input feature provides significant advantages over other input characteristics. The use of JBC as input in the three-stream configuration leads to improved performance. The proposed temporal convolution block, which combines different convolutional operations, shows superior performance compared to individual convolutional blocks. The four-stream configuration achieves the highest evaluating accuracy. Weak position compared to SOTA methods: TFC-GCN's experimental results on the NTU RGB+D60 and NTU RGB+D120 benchmarks are in a weaker position compared to other state-of-the-art (SOTA) methods. It does not achieve the highest accuracy on these datasets. While TFC-GCN performs well on the CSv1 benchmark of the UAV-Human dataset, it has room for improvement on the CSv2 benchmark. Further enhancements are needed to achieve better results on this dataset. Although TFC-GCN has a smaller number of parameters and lower FLOPs compared to some SOTA methods, the experimental results do not provide a comprehensive comparison on these aspects for all methods. The comparison is limited to EfficientGCN-B0.

Author Response

Thanks for your valuable comments. All supplementary contents are highlighted in green in the modified manuscript with annotations. In addition, we will provide answers to the questions you mentioned:

(1). The description of the methodology is brief and lacks in-depth explanations or justifications for design choices. More detailed explanations, including the underlying rationale and theoretical foundations, would enhance the understanding of the proposed approach.

Response: For the description of the method, we have expanded it in the modified manuscript. Please see section 3.

(2). TFC-GCN's experimental results on the NTU RGB+D60 and NTU RGB+D120 benchmarks are in a weaker position compared to other state-of-the-art (SOTA) methods. It does not achieve the highest accuracy on these datasets. While TFC-GCN performs well on the CSv1 benchmark of the UAV-Human dataset, it has room for improvement on the CSv2 benchmark. Further enhancements are needed to achieve better results on this dataset.

Response: As you said, there is a weakness in evaluating accuracy on the three datasets NTU RGB+D, NTU RGB+D 120, and UAV-Human. However, there is a significant advantage in the number of parameters, which is one of our purposes. We will continue to try to improve the evaluating accuracy of the model with such a low number of parameters.

(3). Although TFC-GCN has a smaller number of parameters and lower FLOPs compared to some SOTA methods, the experimental results do not provide a comprehensive comparison on these aspects for all methods. The comparison is limited to EfficientGCN-B0.

Response: We have added more comparison with some models. Please see Table 1 and Table 2. The lightweight models we have listed are already popular as we know them at present.

Reviewer 2 Report

The authors proposed a lightweight TFC-GCN for action recognition. The proposed model contains fewer parameters and shows promising results compared to other state-of-the-art methods. The paper is written nicely and explains the concepts clearly. Moreover, a good comparison is made with other methods on three datasets. Following are the comments that must be incorporated in the revised version.

1.     Line 21: The superiority of the proposed method should be defined in terms of accuracy or computational complexity.

2.     Equation (11): Line 346: Theta represents the lightweight value proposed by the authors. But in reality, there is a tradeoff between accuracy and number of parameters. To solve this tradeoff, the user can define different weights for accuracy and size.

3.     Table 1: X-view and X-Sub in percentage should be defined. How is this percentage calculated?

English is good

Author Response

Thanks for your valuable comments. All supplementary contents are highlighted in red in the modified manuscript with annotations. We will provide a point-by-point answer to the questions you mentioned:

(1). Line 21: The superiority of the proposed method should be defined in terms of accuracy or computational complexity.

Response: We've expanded the definition of strengths on line 21 of the modified manuscript. Please see Lines 21-23, Page 1.

(2). Equation (11): Line 346: Theta represents the lightweight value proposed by the authors. But in reality, there is a tradeoff between accuracy and number of parameters. To solve this tradeoff, the user can define different weights for accuracy and size.

Response: We think the point you put forward is very reasonable. In fact, in the original version, we set a feasible difference interval for the theta of each model, and for all models in this interval, we would choose the most accurate result as the optimal model. For example, if the accuracy of model A is 88.9% and the number of parameters is 1.4M, the calculation of theta is 4.15. The accuracy of model B is 90.5%, the number of parameters is 1.8M, and the calculated theta is 3.91. If we set the feasible difference interval for theta to (-0.3, +0.3), then for models A and B, we will consider model B to be the model we want, even if B has a slightly higher number of parameters than A.

In addition, we have tried to use the Sigmoid function to dynamically adjust the weights in the theta calculation formula, so that the negative evaluation caused by too many parameters can be slightly reduced when the accuracy is very high.

However, in all current studies, both the accuracy and the number of parameters has been maintained in relatively stable numerical intervals. So, in order to simplify the representation of theta, in this modified manuscript, we have chosen to simplify the calculation process, which is the Equation 11:

                                                    θ = ln((α/ρ)+1)                                      (11) 

At present, the number of parameters of mainstream models is within 50M, so there is no need to worry about parameter overflow.

(3). Table 1: X-view and X-Sub in percentage should be defined. How is this percentage calculated?

Response: The evaluating accuracy is calculated as the proportion of correct classification results to all classifications. The definitions of X-View and X-Sub percentages have been supplemented in the modified manuscript. It is expressed as follows in a formula: 

                                    Acc. = (TP+TN)/(TP+TN+FP+FN)

Reviewer 3 Report

The topic could be relevant. There is new contribution from the paper. The application is interesting. Please explain relevance to the focus of journal more explicitly, such as what sensors are used, what sensor data are collected and analysed.

In general, this manuscript is meaningful. However, there are some problems to be solved and areas in need of more explanations.

The Introduction section and Related Work section could be strengthened. The authors are suggested to clarify the research problems, weaknesses of more related works, and to explain the necessity, innovation and comparative advantages of the proposed solution more clearly.

There is only one 2023 reference. Please discuss more recent relevant work.

Please clarify how own solution is better than more recent relevant solutions. 

Please explain the choice of the evaluation criteria, such as, explain with technical details why selected criteria are important for this paper’s application. 

Please explain how the parameters were selected, and how their initial values were assigned.

Explain in good technical details on complexity of the solutions.

Please discuss the weakness and limitation of the proposed solution, and the potential future works.

In page 3, is Fig 1 own innovation? If not, please explain source.

In page 4, is Fig 2 own innovation? If not, please explain source.

In page 7, is Fig 4 own innovation? If not, please explain source.

In page 8, is Fig 5 own innovation? If not, please explain source.

In page 10, is Fig 6 own innovation? If not, please explain source.

In page 11, is Fig 7 own innovation? If not, please explain source.

In page 15, fonts in Figure 8 are too small to read.

For all equations, diagrams and tables, the insights associated with them should be explained.

The language is fine. 

Author Response

Thanks for your valuable comments. All supplementary contents are highlighted in blue in the modified manuscript with annotations. We will provide a point-by-point answer to the questions you mentioned:

(1). Please explain relevance to the focus of journal more explicitly, such as what sensors are used, what sensor data are collected and analysed.

Response: The direction of this article belongs to deep learning, which is one of the receiving directions of the Intelligent Sensors. The focus of our manuscript is on the application of deep learning model based on the sensor data, such as extracting from the different kinds of cameras.

The human body videos we used with the NTU RGB-D60 and NTU RGB-D120 datasets were captured simultaneously by three Microsoft Kinect v2 cameras. Both datasets are from Nanyang Technological University.

Microsoft Kinect v1 is Microsoft's XBOX360 somatosensory peripheral officially announced at the E3 exhibition on June 2, 2009.

At the Xbox One launch event on May 28, 2013, Microsoft showed off the Kinect v2. Compared with the previous generation, this generation of depth images not only has improved resolution, but also has better quality of depth images. The viewable area angle has also been greatly increased. This generation supports up to six people, while the previous generation was only able to identify two. The number of human joints has also increased from 20 in the previous generation to 25. Kinect V2 belongs to the TOF (Time Of Flight) depth camera, intuitively understood to measure the flight time of light to obtain distance; The previous generation used a light coding technology based on the idea of structured light.

Kinect V2 has three coordinate spaces: Color Space, Depth Space; Camera Space。 Among them, the color image is a color space coordinate system; The depth spatial coordinate system is used, in addition to the depth image, it also contains an infrared image and an image of the operator's identification. Both the depth space coordinate system and the color space coordinate system are 2D, that is, the general image coordinate system: the upper left corner is the origin, the right is the positive direction of the X axis, and the down is the positive direction of the Y axis, the unit is pixels, and there is no corresponding to the real length unit. The camera spatial coordinate system is a 3D spatial coordinate system with the sensor as the origin, which is a realistic coordinate system, mainly used for the real 3D environment.

 The human joint points in Kinect V2 human skeleton tracking are recorded in three-dimensional space using the camera spatial coordinate system. Therefore, to draw a human skeleton in OpenCV, you need to first map the 3D coordinates (camera coordinate system) to the 2D coordinates color coordinate system or depth coordinate system.

The UAV-Human dataset was jointly completed by the Singapore University of Technology and Design and Shandong University.

The sensors used are Azure Kinect DK, fisheye camera, and night vision camera. Among them, Azure Kinect DK is a depth camera produced by Microsoft, which is mainly used for human pose capture, environment perception and 3D modeling and other applications. It uses ToF technology to obtain depth information, and also includes a variety of sensors such as RGB image sensors, microphone arrays, and inertial measurement units.

Azure Kinect DK's depth camera uses ToF technology, paired with a high-speed infrared light source and a receiver. A high-speed infrared light source sends a beam of infrared light, and the receiver then measures the time it takes for the light to reach the object and return to the camera, calculating the distance from the object to the camera. The camera has a resolution of 1024x1024 and can acquire depth information within a range of 4 meters. In addition to depth information, the Azure Kinect DK includes an RGB image sensor that can acquire color images. The RGB image sensor has a resolution of 3840x2160 and can capture high-quality images. In addition, the Azure Kinect DK is equipped with a set of microphone arrays and inertial measurement units to capture sound and motion information, enabling more use cases.

The more detailed illustration on the cameras could be found in the following references.

NTU RGB+D [1], NTU RGB+D 120 [2], and UAV-Human [3]. The cites are as follows:

[1] A. Shahroudy, J. Liu, T. Ng, and G. Wang. NTU RGB+D: A large scale dataset for 3D human activity analysis. IEEE Conference on Computer Vision and Pattern Recognition (CVPR), 2016 1010-1019.

[2] J. Liu, A. Shahroudy, M. Perez, G. Wang, L. Duan, and A. Kot. NTU RGB+D 120: A large-scale benchmark for 3d human activity understanding. IEEE Transactions on Pattern Analysis and Machine Intelligence, 2019, 42(10), 2684-2701.

[3] T. Li, J. Liu, W. Zhang, Y. Ni, W. Wang, and Z. Li. UAV-Human: A large benchmark for human behavior understanding with unmanned aerial vehicles. IEEE/CVF Conference on Computer Vision and Pattern Recognition, 2021, 16266-16275.

(2). The Introduction section and Related Work section could be strengthened. The authors are suggested to clarify the research problems, weaknesses of more related works, and to explain the necessity, innovation and comparative advantages of the proposed solution more clearly.

Response: We have revised the introduction and related work sections, and more clearly illustrated the necessity, innovation and comparative advantages of the proposed solutions. Please see section 1 and section 2.

(3). There is only one 2023 reference. Please discuss more recent relevant work.

Response: We introduce some 2023 action recognition models, please see refs. [25][44][50] in the modified manuscript. However, as far as we know, there are still few relevant studies published in 2023, and most of the current excellent models were still released in 2022.

(4). Please explain the choice of the evaluation criteria, such as, explain with technical details why selected criteria are important for this paper’s application. Please explain how the parameters were selected, and how their initial values were assigned.

Response: We chose to measure the parameter theta mainly because we wanted to get a model with the highest possible evaluating accuracy and the lowest possible number of parameters. Therefore, the calculation method with evaluating accuracy as the numerator and parameter quantity as the denominator is chosen to associate the two values. The reason for logarithmic fractions is to compress numeric values.

(5). Please clarify how own solution is better than more recent relevant solutions. Explain in good technical details on complexity of the solutions. Please discuss the weakness and limitation of the proposed solution, and the potential future works.

Response: Compared to other recent relevant solutions, our model has a small number of parameters. This means that our models are easier to be used for real-time computation and deployment on mobile devices than complex, large-parameter models. Our proposed TFC-GCN still has shortcomings. The main weakness is that the results of the model on each dataset are slightly lower, and the efficiency of the model still has room for improvement. Therefore, in our future research, we will start to greatly improve the evaluating accuracy of the model while maintaining the existing number of parameters, and try to deploy it on mobile devices (such as drones) to verify the value of the model.

(6). In page 3, is Fig 1 own innovation? If not, please explain source.

       In page 4, is Fig 2 own innovation? If not, please explain source.

       In page 7, is Fig 4 own innovation? If not, please explain source.

       In page 8, is Fig 5 own innovation? If not, please explain source.

       In page 10, is Fig 6 own innovation? If not, please explain source.

       In page 11, is Fig 7 own innovation? If not, please explain source.

       In page 15, fonts in Figure 8 are too small to read

Response: All the diagrams in this article are drawn by ourselves and the structural of the blocks in the figures are designed by us. The code for our experiment is based on EfficientGCN. For Figure 8, I have the same opinion as you. However, there are 60 categories for the dataset NTU-RGB+D 60, which are all shown in the confusion matrix. This makes the fonts in the Figure 8 impossible to be large enough. Fortunately, the images in our manuscript have a high resolution. Figure 8 in the electronic edition of our manuscript could be zoomed in conveniently.

(7). For all equations, diagrams and tables, the insights associated with them should be explained.

Response: We have further explained all the equations, graphs and tables in the modified manuscript.

Round 2

Reviewer 1 Report

Paper is revised well and shall be accepted in present form.

English is fine to some extent but some minor language corrections shall be carried out.